# Oleacein and Foam Cell Formation in Human Monocyte-Derived Macrophages: A Potential Strategy against Early and Advanced Atherosclerotic Lesions

**DOI:** 10.3390/ph13040064

**Published:** 2020-04-09

**Authors:** Agnieszka Filipek, Tomasz P. Mikołajczyk, Tomasz J. Guzik, Marek Naruszewicz

**Affiliations:** 1Department of Pharmacognosy and Molecular Basis of Phytotherapy, Faculty of Pharmacy, Medical University of Warsaw, Banacha 1, 02-097 Warsaw, Poland; marek.naruszewicz47@gmail.com; 2Institute of Infection, Immunity and Inflammation, University of Glasgow, Sir Graeme Davies Building 120 University Place, Glasgow G12 8TA, UK; tomasz.mikolajczyk@glasgow.ac.uk; 3Department of Internal and Agricultural Medicine, Jagiellonian University Medical College, 31-007 Krakow, Poland; Tomasz.Guzik@glasgow.ac.uk; 4Institute of Cardiovascular and Medical Sciences, University of Glasgow, BHF Centre for Excellence, 120 University Place, Glasgow G12 8TA, UK

**Keywords:** oleacein, human monocyte-derived macrophages (hMDMs), foam cell formation, scavenger receptors (SR), JAK/STAT3 pathway, atherosclerosis

## Abstract

Background: Oleacein is a secoiridoid group polyphenol found mostly in *Olea europea* L. and *Ligustrum vulgare* L. (Oleaceae). The aim of the present study was to investigate a potential role of oleacein in prevention of the foam cell formation. Materials and Methods: Oleacein was isolated from *Ligustrum vulgare* leaves. Human monocyte-derived macrophages were obtained from monocytes cultured with Granulocyte-macrophage colony-stimulating factor (GM-CSF). Then, cells were incubated with 20 μM or 50 μM of oleacein and with oxidized low-density lipoprotein (oxLDL) (50 μg/mL). Visualization of lipid deposition within macrophages was carried out using Oil-Red-O. Expression of CD36, Scavenger receptor A1 (SRA1) and Lectin-like oxidized low-density lipoprotein receptor 1 (LOX-1) was determined by Reverse transcription polymerase chain reaction (RT-PCR) and by flow cytometry. Apoptosis was determined by flow cytometry using Annexin V assay. STAT3 and Acyl-coenzyme A: cholesterol acyltransferase type 1 (ACAT1) levels were determined by ELISA. P-STAT3, P-JAK1, P-JAK2 expressions were determined by Western blot (WB). Results: Oleacein in dose-dependent manner significantly reduced lipid deposits in macrophages as well as their expression of selected scavenger receptors. The highest decrease of expression was found for CD36 and SRA1 receptors, from above 20% to more than 75% compared to oxLDL and the lowest for LOX-1 receptor, from approx. 8% to approx. 25% compared to oxLDL-stimulated macrophages. Oleacein significantly reduced (2.5-fold) early apoptosis of oxLDL-stimulated macrophages. Moreover, oleacein significantly increased the protein expression of JAK/STAT3 pathway and had no effect on ACAT1 level. Conclusions: Our study demonstrates, for the first time, that oleacein inhibits foam cell formation in human monocyte-derived macrophages and thus can be a valuable tool in the prevention of early and advanced atherosclerotic lesions.

## 1. Introduction

Atherosclerosis is chronic inflammatory vascular disease which leads to stroke and myocardial infarction and is therefore a primary cause of cardiovascular death. The pathogenesis of atherosclerosis involves a complex interplay between monocytes, macrophages and low-density lipoproteins (LDL), principally oxidized LDL (oxLDL). Endothelial cell (EC) dysfunction caused by oxLDL initiates a series of pro-inflammatory reactions, which result in activation of adhesive molecules (intercellular adhesion molecule 1 (ICAM-1), vascular cell adhesion molecule 1 (VCAM-1)) and chemokines (macrophage colony-stimulating factor (M-CSF), monocyte chemo attractant protein-1 (MCP-1)). This leads to an increased migration of monocytes into the vascular intima. There, monocytes transform into macrophages, which take in oxLDL and transform into foam cells. Foam cells are therefore a basic component of atherosclerotic plaque and play an important role in development of early and late atherosclerotic lesions [1].

oxLDL interaction with macrophages precedes the uptake process and is mediated by scavenger receptors (SRs) on the macrophage surface. SRs can bind lipoproteins, polyanions, and cells in apoptotic or necrotic phases. SRs are categorized into seven classes according to their structural characteristics, ranging from SR-A to SR-G. So far, it has been discovered that three scavenger receptors, SRA1 (class A), CD36 (class B) and LOX-1 (lectin-like oxLDL receptor; class E), which were identified on macrophages, are primary receptors involved in binding and uptake of oxidized low-density lipoproteins [2,3,4]. Despite numerous attempts, effective cardiovascular therapies targeting these mechanisms are yet to be developed. Here, we propose that oleacein (3,4-DHPEA-EDA, the dialdehydic form of decaboxymethylelenolic acid linked to hydroxytyrosol, Figure 1), a polyphenol derived from the secoiridoids group, may potently inhibit foam cell formation.

Oleacein is found primarily in plants from the *Oleacea* family, (i.e., *Olea europea* L. and *Ligustrum vulgare* L). Oleacein has been shown to target a number of pro-atherosclerotic mechanisms. It has been shown to inhibit ROS production, myeloperoxidase secretion and CD11b/18 expression in human neutrophils [5,6,7]. Moreover, oleacein prevents H_2_O_2_-induced DNA damage in monocytes, and inhibits expression of adhesion molecules such as VCAM-1, ICAM-1 and E-selectin, consequently reducing monocyte adhesion to Human umbilical vein endothelial cells(HUVEC) [5,8,9]. Oleacein protects endothelial progenitor cells (EPCs) against the pathogenic effects of Ang II via activation of the Nrf2/HO-1 pathway and restores the neovascularization and angiogenesis ability of EPCs [10].

Most importantly, we have demonstrated a potential influence of oleacein on stabilization of human atherosclerotic plaque. Our results suggest that oleacein enhances the anti-inflammatory activity of complex hemoglobin with haptoglobin 1-1 and 2-2. Moreover, oleacein increases the expression of CD163 and IL-10 receptors, as well as HO-1 secretion, and may change cell phenotype of macrophages from pro-inflammatory (M1) to anti-inflammatory (M2) [11]. Switching the phenotype of macrophages is likely to be important in stabilization of atherosclerotic plaques in human arteries. In ex vivo studies (carotid plaques were obtained during endarterectomy of 20 patients in age 50–79 years with TIA lasting less than 24 h), it has been confirmed that oleacein may attenuate the destabilization of carotid plaque by decreasing expressions of high mobility group box 1 protein (HMGB1), matrix metallopeptidase 9 (MMP-9), matrix metallo proteinase-9/neutrophil gelatinase-associated lipocalin (MMP-9/NGAL complex) and tissue factor (TF) secretion [12].

As these studies indicate that oleacein may target several key pathomechanisms of atherosclerosis, it is essential to understand its effects on foam cell formation as a critical step in atherosclerotic plaque formation and stabilization.

## 2. Materials and Methods

### 2.1. Oleacein

Oleacein (OC) was isolated from *Ligustrum vulgare* L. (*Oleaceae*) leaves as previously described [13]. The structure and purity of compound was confirmed by UV, NMR and MS spectra. The purity of compound was confirmed to be >95% through TLC and HPLC methods. The plant material is described in the Appendix A. Oleacein was dissolved in DMSO (Sigma Life Science, Darmstadt, Germany) and then in (Ca^2+^/Mg^2+^)-free phosphate buffered saline (PBS) at pH 7.4 to a final concentration of 20 μM (OC_20_) and 50 μM (OC_50_). The final concentration of DMSO did not exceed 0.01% and did not influence the performed assays.

### 2.2. Blood Collection

Peripheral blood samples and autologous serum were obtained from healthy adult volunteers under 35 years of age, from the Warsaw Donate Blood Centre. Based on their medical history and routine laboratory tests, donors were determined as being healthy. They declared themselves as not being smokers, or as currently taking any form of medication. The study conformed to the principles of the Declaration of Helsinki.

### 2.3. Isolation and Cultivation of Macrophages

Immediately after collection, 9 mL heparinized blood was subjected to a 2-fold dilution with PBS. Next, the blood was layered over 9 mL of Lymphocyte Separation Medium (LSM 1077, PAA, Laboratories GmbH, Pasching, Austria) and centrifuged (400× *g*, 20 min, 4 °C). The peripheral blood mononuclear cells (PBMCs) were collected by aspiration, then washed with cold PBS (PAA, Laboratories GmbH), and centrifuged. The cells were suspended in RPMI 1640 medium with L-glutamine and HEPES (PAA, Laboratories GmbH), antibiotics: penicillin (100 U/mL), streptomycin (100 μg/mL), amphotericin and gentamycin (2.5 μg/mL) (Sigma Aldrich Chemie GmbH, Steinheim, Germany), and heat inactivated autologous serum (20%). To allow the adherence of monocytes, the peripheral blood mononuclear cells suspension was placed in 12-well tissue culture plates (2 × 10^6^/well) and incubated for 2 h at 37 °C under humidified 5% CO_2_ air. After that, non-adherent cells were removed and adherent cells were cultured with granulocyte macrophage-colony stimulating factor (GM-CSF, 10 ng/mL; Sigma Aldrich Chemie GmbH) for 7 days to induce differentiation to macrophages [14,15]. The medium supplemented with autologous serum was replaced every 2 or 3 days.

### 2.4. Foam Cell Formation

The macrophages were incubated with or without oleacein (20 μM and 50 μM) for 1 h followed by incubation with oxLDL at a concentration 50 μg/mL (Thermo Fisher Scientific, Waltham, Germany) for 72 h [16]. Oil-Red-O (Sigma Aldrich Chemie GmbH) was used to visualize lipid deposition in macrophages [17]. Positive-staining cells were macrophage-derived foam cells, which observed via fluorescence microscope (TS100F, Nikon, Tokio, Japan) and then photographed using Image software (NIS-Elements BR, Melville, NY, USA. The number of cells was counted using Bürker Counting Chambers (Göteborg—Sweden). The results were reported as a percent of cells with lipid deposits in comparison to oxLDL-stimulated macrophages (100%).

### 2.5. The Expression of CD36, SRA1 and LOX-1 Receptor

The macrophages were incubated with or without oleacein (20 μM and 50 μM) for 1 h followed by incubation with oxLDL (50 μg/mL) for 24 h. The expression of CD36 mRNA, SRA1 mRNA and LOX-1 mRNA in macrophages was determined by western blot. Total RNA was obtained from cells using a RNeasy Mini Kit (Qiagen, Valencia, CA, USA)). Reverse transcription of 1 μg RNA was performed using High Capacity cDNA Reverse Transcription Kit (Applied Biosystems, Foster City CA, USA)). mRNA expression of CD36 (Hs00354519_m1), SRA (Hs00234007_m1), LOX-1 (Hs01552593_m1) genes were analysed using TaqMan^®^ probes (Thermo Fisher Scientific) and TaqMan^®^ Real-Time PCR Master Mix (Thermo Fisher Scientific). Reactions were prepared and run on 384-well plates on the QuantStudio^TM^ 12K Flex Real-Time PCR System with standard protocol. Calculations were made using QuantStudioTM Real-Time PCR Software. Data were normalized to levels of Glyceraldehyde-3-phosphate dehydrogenase (GAPDH) (Hs02786624_g1) mRNA and relative quantification was calculated as 2^−ΔΔCt^.

Effect of oleacein on CD36, SRA1 and LOX-1 expression has been confirmed using flow cytometry FACSCalibur (BD Biosciences, San Jose, CA, USA). Cells were incubated for 30 min with fluorescently labelled monoclonal antibodies anti-CD36-PE (clone CB38, BD Pharmingen, San Jose, CA, USA), anti-MSR1 (SRA1)- PE (clone U23-56, BD Biosciences) and anti-LOX-1 –FITC (clone 23C11, Abcam, Cambridge, UK) according to manufacturer’s recommendations. The following isotope controls were used: PE mouse IgG1, k Isotype Control (for SR-A1), PE Mouse IgM, k Isotype Control (for CD36) and FITC Mouse IgG1, k Isotype Control (for LOX-1) (BD Pharmingen). The mean fluorescence intensity (MFI) in the gated cell population was measured in the fluorescence FL2 or FL1 channels (10,000 cells per sample). The results were expressed as the number of cells expressing the CD36, SRA1 or LOX-1 receptor, compared to cells stimulated with oxLDL.

### 2.6. Apoptosis Assay by Flow Cytometry Analysis

The macrophage cells were incubated with oxLDL (100 μg/mL) or oxLDL with oleacein (20 μM and 50 μM) for 24 h. Annexin V and PI staining (BD Pharmingen) were performed to detect early stage of apoptosis in macrophages according to the manufacturer’s instruction. The cells (1 × 105) were suspended in Annexin V Binding Buffer (100 μL) and then mixed with 5 μL of annexin V^−^ Fluorescein isothiocyanate (FITC) and 10 μL of propidium iodide (PI). After incubation (15 min) in the dark at room temperature, the samples were analysed by flow cytometry. The number of early apoptotic cells (Annexin V^+^/PI^−^) was determined.

### 2.7. STAT3 Cellular Level and ACAT1 Expression

The cellular level of STAT3 and expression of ACAT1 in macrophages was measured by enzyme-linked immunosorbent assay (Elisa Kit for Signal Transducer and Activator of Transcription 3, Cloud-Clone Corp., Katy, TX, USA), Human ACAT1 (Sterol-O-Acyltransferase 1 (SOAT1) ELISA Kit, MyBioSource, San Diego, CA, USA) according the protocol provided by the manufacturer. After the incubation of cells with or without oleacein (20 μM and 50 μM) for 1 h and then incubation with oxLDL (50 μg/mL) for 72 h, macrophages were washed in cold PBS gently, and then detached with cell dissociation solution non-enzymatic (Sigma Aldrich Chemie GmbH), and collected by centrifugation at 1000× *g* for 5 min. Cells were washed 3-times in cold PBS. Then, macrophages were resuspended in fresh cell lysis buffer 1 (R&D Systems, a Biotechne Brand, Minneapolis, MN, USA) and centrifugated at 1500× *g* for 10 min at 4 °C. After removed of cellular debris, aliquots were collected and stored at −70 °C for analysis. Protein concentration was quantified by a standard colorimetric test (Pierce^TM^ BCA Protein Assay Kit, Thermo Scientific, Waltham, MA, USA).

### 2.8. The Level of P-STAT3, P-JAK1, P-JAK2

The level of P-STAT3/STAT3, P-JAK1/JAK1 and P-JAK2/JAK2 in macrophages were determined by western blot. After incubation, the macrophages were collected and centrifuged (1500× *g*, 10 min, 4 °C). Cells were lysed in ice-cold lysis buffer with phosphatase and protease inhibitors (Sigma Aldrich Chemie GmbH), and the resulting lysates were centrifuged (800× *g*, 15 min, 4 °C). Protein concentration was quantified by a standard colorimetric test (BCA Protein Assay Kit) and 15 µL of lysate was resuspend in 5 µL Lamelli Buffer (Sigma Aldrich Chemie GmbH), centrifuged, shortly vortex, boiled 5 min in 95 °C, vortexed and frozen as aliquots at −70 °C until analysed by 12% SDS-PAGE. The protein in amount of 40 µg was transferred to nitrocellulose filters and immunoblotted with anti-P-STAT3, anti-STAT3, anti-P-JAK1, anti-JAK1, anti-P-JAK2, anti-JAK2 (Cell Signalling Technology, Beverly, MA, USA) at 1:1000 dilutions and a rabbit anti-actin polyclonal antibody at a 1:1000 dilution. Peroxidase-conjugated AffiniPure goat anti-rabbit antibody was used as a secondary antibody at a dilution of 1:10,000. Finally, the blots were incubated with chemiluminescent substrate for the detection of horseradish peroxidase (HRP) (Thermo-Scientific, USA) for 10–15 min. Western blots were quantified using the Image J 1.38 software (NIH, Bethesda, MD, USA) after densitometric scanning of the bands.

### 2.9. Controls

The solution of pitavastatin (PIT) at a concentration of 20 μM was used as inhibitor of scavenger receptors expression [18]. AZD1480 (Sigma Aldrich Chemie GmbH) at a concentration of 0.5 μM was used as an inhibitor for determination of JAK/STAT3 pathway.

### 2.10. Cytotoxicity Assay

Cytotoxicity of oleacein, oxLDL and oleacein with oxLDL for macrophages was measured using a Cytotoxicity Detection Kit (LDH) (Roche Diagnostics GmbH, Mannheim, Germany) according the protocol provided by the manufacturer.

### 2.11. Statistical Analysis

The results were expressed as a mean ± SEM. One-way analysis of variance (ANOVA) (>3 groups) with Tukey’s post-hoc test or Student’s test (2 groups) was used for statistical analysis. All analyses were performed using STATISTICA software v. 10.0 PL (StatSoft, Tulsa, OK, USA). Values of * *p* < 0.05 and ** *p* < 0.001 were considered significant.

## 3. Results

### 3.1. Effect on Cytotoxicity

After 24 and 72 h of macrophage incubation with oleacein (20 μM, 50 μM) or pitvastatin (20 μM) and oxLDL (50 μg/mL, 100 μg/mL) no cytotoxic effect on the cells was observed (Appendix A).

### 3.2. Effect of Oleacein on oxLDL-induced Foam Cell Formation

It is known that macrophage uptake of oxLDL-forming foam cell is an indication of carotid plaque. Incubation of macrophages with oxLDL (50 μg/mL, 72 h) resulted in lipid deposition and foam cell formation (Figure 2A). The quantitative analysis showed that oleacein, in a dose-dependent manner, significantly inhibits oxLDL-induced foam cell formation.

A decreased number of foam cells was observed for oleacein, by approximately 20% at a concentration of 20 μM, and approximately 80% at a concentration of 50 μM (Figure 2B, *p* < 0.05; *p* < 0.001). Pitavastatin decreased the number of lipid deposition cells by over 40% compared to oxLDL-induced macrophages (Figure 2B, *p* < 0.001).

### 3.3. Effect of Oleacein on CD36, SRA1 and LOX-1 Expression

The decrease of CD36, SRA1 and LOX-1 expression was investigated at the level of mRNA, using quantitative RT-PCR. In line with protein levels, oleacein attenuated oxLDL-induced CD36 mRNA, SRA1 mRNA and LOX-1 mRNA expression. The most pronounced decrease in expression was observed for CD36 mRNA (23% OC20, *p* < 0.05; 84% OC50, *p* < 0.001) and SRA1 mRNA (20% OC20, *p* < 0.05; 65% OC50, *p* < 0.001) (Figure 3A). The decreased LOX-1 mRNA expression was less pronounced and observed only in response to a higher concentration (50 μM) of oleacein, which led to a reduction of 16% (*p* < 0.05) when compared to macrophages stimulated by oxLDL in the absence of oleacein (vehicle only), while oleacein at the concentration of 20 μM also led to a modest decrease in LOX-1 mRNA, this change was not statistically significant (Figure 3A). As a positive control, and as comparison, we used pitavastatin, which caused a pronounced decrease in the expression of all tested scavenger receptors, by approximately 55% (CD36), 44% (SRA1) and 38% (LOX-1) (Figure 3A, *p* < 0.05, *p* < 0.001).

Furthermore, the activity of SRs was confirmed by flow cytometry. Oleacein, in a dose-dependent manner, significantly reduced the expression of all selected scavenger receptors in macrophages stimulated with oxLDL (Figure 2B). The most pronounced reduction of surface expression was observed for CD36. On ox-LDL-treated macrophages (7803 ± 323 number of cells expressing CD36) CD36 expression was reduced by 20% when oleacein was used at a concentration of 20 μM (OC20; 6141 ± 305 cells expressing CD36, *p* < 0.001) and by 80% (OC50; 1285 ± 335 cells expressing CD36, *p* < 0.001) for oleacein at a concentration of 50 μM (OC50, Figure 3B). A statistically significant decrease of the SRA1 receptor expression was observed. On oxLDL-induced macrophages (7147 ± 178 cells expressing SRA1) SRA1 expression was reduced by 20% for OC20 (5498 ± 319 cells expressing SRA1; *p* < 0.05) and by 75% for OC50 (2411 ± 377 cells expressing SRA1; *p* < 0.001, Figure 3B).

The less pronounced decrease in expression was observed for the LOX-1 receptor. A statistically significant reduction in LOX-1 receptor expression was determined only for oleacein for a concentration of 50 μM, by 25% (5977 ± 115 cells expressing LOX-1; *p* < 0.05) vs. oxLDL treated macrophages (8107 ± 317 cells expressing LOX-1). In addition, we observed that OC20 decreased the expression of the LOX-1 receptor by 8% (7422 ± 388 cells expressing LOX-1), but the differences were not significant (Figure 3B). On oxLDL-induced macrophages CD36, SRA1 and LOX-1, expression was reduced by 20% when pitavastatin was used (Figure 3B; *p* < 0.05, *p* < 0.001).

### 3.4. Effect of Oleacein on Apoptosis of oxLDL-Induced Macrophages

It has been shown that apoptotic and necrotic processes of cells are characteristic of instability in carotid plaques in atherosclerotic lesions. The majority of apoptotic cells are macrophages, which are localized near the necrotic core in advanced atherosclerotic plaques [19]. OxLDL significantly increases the number of early apoptotic cells (Annexin V^+^/PI^−^; 15.51 ± 3.26% vs. 1.88 ± 0.37% in control; *p* < 0.001) (Figure 4). Oleacein, at a concentration of 50 μM, statistically significantly reduced (by 2.5-fold) early apoptosis of oxLDL-stimulated cells (6.17 ± 1.79% vs 15.51 ± 3.26% for oxLDL-induced macrophages; *p* < 0.001). Lower values, although not significant, were determined for oleacein at a concentration of 20 μM (10.93 ± 1.71% vs 15.51 ± 3.26% for oxLDL-induced macrophages) (Figure 4A,B). Pitavastatin also reduced early apoptosis, but the differences were not statistically significant (Figure 4A,B).

### 3.5. Effect of Oleacein on STAT3

The role of STAT3 in atherosclerosis is not fully known. However, it is supposed that STAT3 may change the macrophage phenotype from pro-inflammatory M1 to anti-inflammatory M2 [20,21] via activation of the JAK/STAT pathway. Oleacein, in a dose-dependent manner, significantly increased STAT3 cellular level from 2.68 ± 0.87 ng/g cellular protein (oleacein at a concentration of 20 μM) to 4.82 ± 0.59 ng/g cellular protein (oleacein at a concentration of 50 μM) compared to the oxLDL-stimulated macrophages (0.68 ± 0.17 ng/g cellular protein) (Figure 5A, *p* < 0.05, *p* < 0.001). Pitavastatin also increased STAT3 cellular level but the differences were not statistically significant (Figure 5A).

In non-stimulated cells, STAT3 is kept in an inactive cytoplasmic form. After activated, STAT3 translocate into the nucleus where it behaves as a transcription activator for a broad array of targeted genes. STAT3 activation is induced by phosphorylation of tyrosine. Phosphorylated STAT3 (P-STAT3) is active in both normal physiological and pathological conditions [22]. The expression of P-STAT3 was determined by western blot. Oleacein significantly increased the P-STAT3 expression by four times for OC20 (3.86 ± 0.51, *p* < 0.001) and five times for OC50 (5.00 ± 0.43, *p* < 0.001) compared to the oxLDL-induced macrophages (Figure 5B). After treatment with pitavastatin on oxLDL-induced macrophages, we did not observe any change in P-STAT3 expression (Figure 5B). STAT3 pathway inhibitor, AZD1480, decreased the expression of P-STAT3 for all treatment factors compared to factors without AZD1480 (Figure 5B; *p* < 0.001).

### 3.6. Effect of Oleacein on JAK1, JAK2 Expression

It is known that the change in macrophage phenotype from M1 to M2 is associated with the activation of the JAK/STAT3 pathway [23]. We observed that oleacein at a concentration 20 μM and 50 μM significantly increased the P-JAK1 expression by three times (2.94 ± 0.47; 3.31 ± 0.68; *p* < 0.001) compared to oxLDL-induced macrophages (Figure 5B). The expression of P-JAK2 was intensified by oleacein with a concentration of 20 μM and 50 μM, 2.55 ± 0.33 (*p* < 0.001) and 3.33 ± 0.67 (*p* < 0.001), respectively. After the use of pitavastatin, the expression of P-JAK1 and P-JAK2 also increased, but the difference was not statistically significant (Figure 5B). Using JAK1 and JAK2 pathway inhibitor, AZD1480 was greatly decreased the expression of P-JAK1 and P-JAK2 for all treatment factors compared to the factors without AZD1480 (Figure 5B; *p* < 0.001).

### 3.7. Effect of Oleacein on ACAT1

Acyl-coenzyme A: cholesterol acyltransferase 1 (ACAT1) plays a major role in cholesterol homeostasis. It is an intracellular enzyme that converts free cholesterol into cholesteryl esters for storage in lipid droplets and promotes foam cell formation in atherosclerotic lesions [24]. We examined whether oleacein has an effect on ACAT1 activity. After treatment with oleacein (20 μM and 50 μM) or pitavastatin on oxLDL-induced macrophages, we did not observe any change in ACAT1 expression (Appendix A).

## 4. Discussion

Our results show for the first time that oleacein may attenuate foam cell formation. This effect has been related to decreased expression of macrophages receptors, such as SRA, CD36 and LOX-1. Moreover, in an animal model, oleacein may have attenuated foam cell formation in the livers of mice [25].

It is well known that several scavenger receptors are involved in foam cell formation via uptake of oxidized low-density lipoproteins. The most important functions are attributed to the CD36, SRA1 and LOX-1 receptors in foam cell formation and this increases the risk of development of atherogenesis [4].

The CD36 receptor (88 kDa) also known as FAT/CD36, (FAT)/CD36, SCARB3 and GP88, is a heavily glycosylated transmembrane protein that belongs to SR class B family. It consists of an extracellular domain flanked by two membranes and two cytoplasmic domains. The main function of the CD36 receptor is a high affinity for oxLDL. It is well known that CD36 expression is regulated by multiple factors. Oxygenated carotenoid and palmitate significantly increase the expression of the CD36 receptor. A lipophilic diterpene isolated from *Salvia miltiorrhiza* decreases the expression levels of CD36. Similarly, fatty acid and antioxidant activity of walnut or olive oil is associated with reduced CD36 expression [3,16,26]. SRA, a 77kDa cell surface glycoprotein, is a member of the class A scavenger receptor family. The human SRA is located on chromosome 8 and can be transcribed to produce three splice variants: SRA1, SRA2 and SRA3. SRA is highly expressed on macrophages and mediates the uptake of oxLDL by macrophages. Inhibition of SRA in macrophages significantly reduces foam cell formation in apoE^−/−^ mice. Moreover, it is considered that some compounds, such as kaempferol, hydrogen sulfide, curcumin and polyphenolic extracts from mulberry leaves, decrease SRA expression [2,27,28].

LOX-1 (50 kDa) was initially identified as the major receptor for oxLDL in endothelial cells. Recently it has been discovered in macrophages and vascular smooth muscle cells in artery vessels. The interaction of LOX-1 and oxLDL induces several processes such as endothelial dysfunction, macrophage-derived foam cell formation, leukocyte adhesion. Current studies have proven, that LOX-1 activation eventually leads to the rupture of atherosclerotic plaques and acute cardiovascular events [29]. LOX-1 is a type II membrane protein belonging to the C-type lectin family of molecules and classified as an E class scavenger receptor. LOX-1 is structurally different from other known oxLDL receptors, including class A and class B, and consists of four domains: a short N-terminal cytoplasmic domain, a transmembrane domain, a connecting neck domain and a lectin-like domain at the C-terminus. LOX-1 expression is induced by pro-inflammatory factors, such as homocysteine, PMA, TNFα and inflammatory cytokines, whereas the statins superoxide dismutase or PPARγ activator decrease LOX-1 expression [30,31,32]. It is known, that LOX-1 is absent in human monocytes but it is activated in macrophages with oxLDL, lysophosphatidylcholine (LPC), palmitic acids and glucose. Moreover, LOX-1 does not alter the uptake of oxLDL in unstimulated macrophages. The expression of LOX-1is induced by pro-inflammatory cytokines, and if that occurred in macrophages in atherosclerosis, LOX-1 increased oxLDL uptake by lesioned macrophages [33].

It is considered that the CD36 receptor plays a critical role in the initiation of atherosclerotic lesions by its ability to bind and internalize modified LDL in the artery wall. The CD36 receptor takes in over 75% of oxLDL, and stimulates macrophage foam cell formation and the release of inflammatory cytokines [26]. Moreover, in vitro studies have shown that scavenger receptors such as SRA and LOX-1 are relevant for cholesterol uptake [3]. Our study demonstrated that oleacein dose-dependently inhibited expression of CD36, SRA1 and LOX-1. However, the highest decrease in expression was noted in the case of the CD36 and SRA1 receptors. The lowest decrease in expression was observed in the LOX-1 receptor. Our results may suggest that oleacein can attenuate uptake of modified cholesterol by macrophages and can inhibit foam cell formation, thus preventing early atherosclerotic lesions. Moreover, it is known that oleacein metabolites such as apigenin-7-O-glucuronide may inhibit CD36 expression, resulting in an anti-atherosclerotic effect [34].

Chemically oleacein is a polyphenol. Our studies show different directions of action of oleacein. However, other phenolic compounds with similar effects are also known. Ellagic acid, a polyphenolic compound found in fruit (raspberries, blueberries, strawberries, grapes, pomegranates) and nuts, may protect against oxLDL-induced endothelial dysfunction by modulating the signaling pathway via LOX-1 [35]. Then, resveratrol (polyphenol derivative of stilbene) attenuates lipid accumulation in cultured human macrophages via effects on cholesterol transport [36]. In addition, it has been observed that extracts rich in polyphenolic fraction from mulberry leaves, as well as pomegranate juice and peel can lead to reduced cellular cholesterol accumulation in macrophages and foam cell formation [37,38].

Pitavastatin is a novel inhibitor of HMG-CoA reductase. Clinical trials have shown that pitavastatin decreases serum LDL cholesterol, triglycerides level and increases HDL cholesterol. It is known that pitavastatin downregulates the expression of CD36 receptor. In addition, pitavastatin reduces oxLDL uptake by macrophages through PPARγ-dependent pathway [18]. Therefore, we suppose that the PPARγ pathway can also be activated by oleacein.

ACAT plays an important role in cholesterol homeostasis by regulating the distribution of cellular cholesterol to free and esterified cholesterol pools. Cholesterol esterification results in its stored in cellular lipid drops and leads to the foam cell formation. Esterified cholesterol is not available for efflux from cells using ABC transporters (e.g., ABCA1) [24]. Our studies have shown that oleacein does not inhibit ACAT1 expression. This data may support the hypothesis that oleacein activity is only associated with the scavenger receptor mechanism. Moreover, our current research confirms that oleacein can potentiate the efflux of cholesterol from macrophages by activating ABC transporters.

Manning-Tobin et al. showed that CD36 and SRA receptors not only initiate early atherosclerotic lesions, but also contribute to macrophage apoptosis and plaque necrosis in advanced lesions [28]. It has been shown that apoptotic macrophages are localized near the necrotic areas of advanced lesions [39]. Macrophage apoptosis is a significant feature of the development of atherosclerotic plaque and occurs throughout atherogenesis. In addition, numerous papers suggest that macrophage apoptosis is associated with unstable atherosclerotic plaques that are prone to rupture, resulting in micro-hemorrhage, pro-coagulating and pro-thrombotic molecule accumulation, and vessel closure [40]. Our study showed that oleacein, at a concentration of 50 μM, inhibited oxLDL-induced early apoptosis of macrophages. Therefore, our results may imply a potential role of oleacein for patients with advanced atherosclerotic lesions.

In atherosclerotic plaques, macrophages change their phenotype in response to various stimuli. Pro-inflammatory macrophages M1 are stimulated by IFN-γ, LPS, modified LDL and cholesterol crystals. Moreover, inhibition of macrophage migration by oxidized phospholipids and embedding them in tissue by oxLDL promotes a pro-inflammatory phenotype [41]. Therefore, M1 macrophages are the predominant population in rupture zones of unstable plaques, whereas anti-inflammatory macrophages M2 are activated by microenvironmental factors such as cytokines (IL-4, IL-10), scavenger receptors (CD163, IL-10), transcription factors (PP) and polyunsaturated fatty acids or sphingosine-1-phoshate [42]. M2 macrophages have anti-hemorrhagic properties and protect against foam cell formation. In our previous study we showed that oleacein can change the phenotype of macrophages from pro-inflammatory M1 to anti-inflammatory M2 by increasing CD163 and IL-10 receptors [11]. In this study, we discovered that the change in the macrophage phenotype by oleacein is dependent on the JAK/STAT3 pathway. Independent of the dose, oleacein increased the intracellular secretion of STAT3. More importantly, oleacein significantly improved the phosphorylation of STAT3, JAK1 and JAK2. Oleacein affected not only phosphorylation level but also protein expression STAT3. It could be due to the proteins stabilization by agents that increase chaperone protein activity [43].

Other studies suggested that polarization of pro-inflammatory macrophages into M2 may happen via the Janus kinase signal transducer and activation of transcription-1 (JAK/STAT1), as well as the STAT3 pathway. Activation of the STAT3 transcription factor is dependent on the autophosphorylation of the IL-10 receptor, which is composed of two heterodimers: IL10R1 and IL10R2. The IL-10 receptor, as well as the JAK-dependent STAT3 transcription factor, mediates the suppression of cytokine proinflammatory signaling pathways [23]. Hutchins et al. confirmed that JAK activation is upregulated by IL-10R expression. IL-10R activates the IL-10/JAK1/STAT3 cascade, where phosphorylated STAT3 translocate to the nucleus to activate the target genes expression [44]. Oleacein strongly induces the expression of IL-10 receptor [11,12]. Therefore, we suggest that the mechanism of activation of the JAK pathway by oleacein involves the IL10R/JAK/STAT3 pathway. It has been shown that macrophage STAT1 stimulates foam cell formation and macrophage apoptosis, whereas activation of the JAK/STAT3 pathway changes the macrophage phenotype from M1 to M2, thereby inhibiting development of both early and advanced atherosclerotic lesions [21,45]. 

It should be taken into account that the role of the JAK/STAT3 pathway is still inconsistent. JAK/STAT3 overexpression may promote the development of atherosclerosis by facilitating inflammation, proliferation, differentiation and migration of vascular cells. Whereas, the inhibitor of the JAK/STAT3 pathway, such as protein inhibitor of activated STAT-3 (PIAS3), may be a critical repressor of atherosclerosis [22]. Moreover, it is known that P-STAT3 plays the role of oncogenic protein and is observed in about 70% of human tumors [46]. Therefore, this problem still requires a detailed explanation.

Our data suggests that oleacein decreases foam cell formation and shifts the polarization towards M2 phenotype however the mechanism is non-fully understood. Nevertheless, current in vivo studies support our hypothesis that oleacein may prevent the accumulation of cholesterol in mouse liver macrophages. [25].

In the discussion on the potential role of oleacein as a medicinal substance, its bioavailability should be taken into account. It is known that about 70% of oleacein reaches the small intestine in unchanged form and then enters the circulatory system in the form of gluconic acid and glucuronides [47,48]. It should be remembered that inflammatory cells may secrete liposomal β-glucuronidase into the extracellular space and oleacein may be re-released from glucuronides in regions of inflammation [49,50].

The best source of oleacein is unfiltered extra virgin olive oil. Unfortunately, we have several restrictions here. The oleacein content in oils is very different and depends on many factors such as olive tree growing conditions, temperature, insolation, hydration and the method of pressing olive oil. The richest in oleacein are olive oils from the Greek islands and polyphenol-rich unfiltered extra virgin olive oil may not be available to consumers other than Mediterranean countries [48,51]. It should be noted, that polyphenols are an unstable compound in higher temperature. For this reason, extra virgin olive oil should be consumed in a raw form [52]. In addition, olive oil is rich in fatty acids. Consumption of more than 2 tablespoons of extra virgin olive oil can exceed the daily requirement of adult for fatty acids. Therefore, the daily intake of polyphenols can be too low to induce a therapeutic effect. Due to the above reason, we recommend using oleacein in the form of a pure compound as a dietary supplement or medical drug.

## 5. Conclusions

Our results suggest that oleacein may reduce foam cell formation from human macrophages by decreasing expression of scavenger receptors such as CD36, SRA1 and LOX-1. Moreover, oleacein can protect against foam cell formation by switch macrophages from pro-inflammatory type M1 to anti-inflammatory type M2 via JAK/STAT3 pathway activation. In addition, oleacein may inhibited oxLDL-induced early apoptosis of macrophages. This beneficial effect is associated with the multifaceted, so-called pleiotropic activity of oleacein. Therefore, we suggest oleacein as a substance for use in strategies against early and advanced atherosclerosis. Noteworthy is the fact that the source of oleacein may be the leaves of *Ligustrum vulgare* (*Oleaceae*), which is a common plant, not only in Mediterranean countries. Nevertheless, these studies still require confirmation on an animal model. However, they indicate that in the future oleacein may play a significant role in preventing heart disease.

## Figures and Tables

**Figure 1 pharmaceuticals-13-00064-f001:**
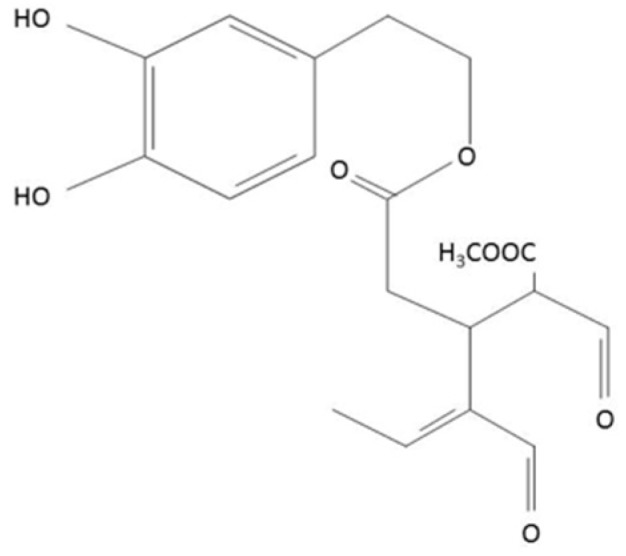
Chemical structure of oleacein.

**Figure 2 pharmaceuticals-13-00064-f002:**
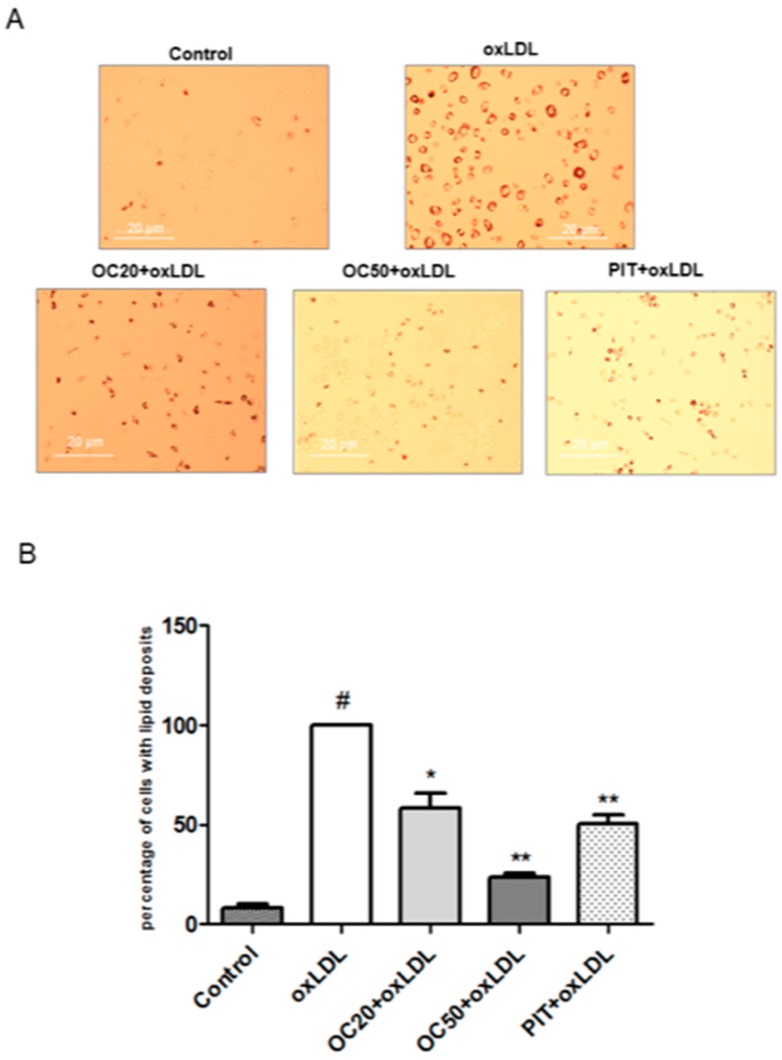
Influence of oleacein on oxLDL-induced foam cell formation in macrophages. The macrophages were incubated with or without oleacein (20 μM and 50 μM) or pitavatatin for 1 h followed by incubation with oxLDL at a concentration of 50 μg/mL for 72 h. (**A**) Oil Red O staining was used to visualize lipid deposition in macrophages (*n* = 12). (**B**) Data is a percentage of cells with lipid deposition compared to the oxLDL-induced macrophages (100%). Statistical significance * *p* < 0.05, ** *p* < 0.001. #—constant relative to which statistics are calculated.

**Figure 3 pharmaceuticals-13-00064-f003:**
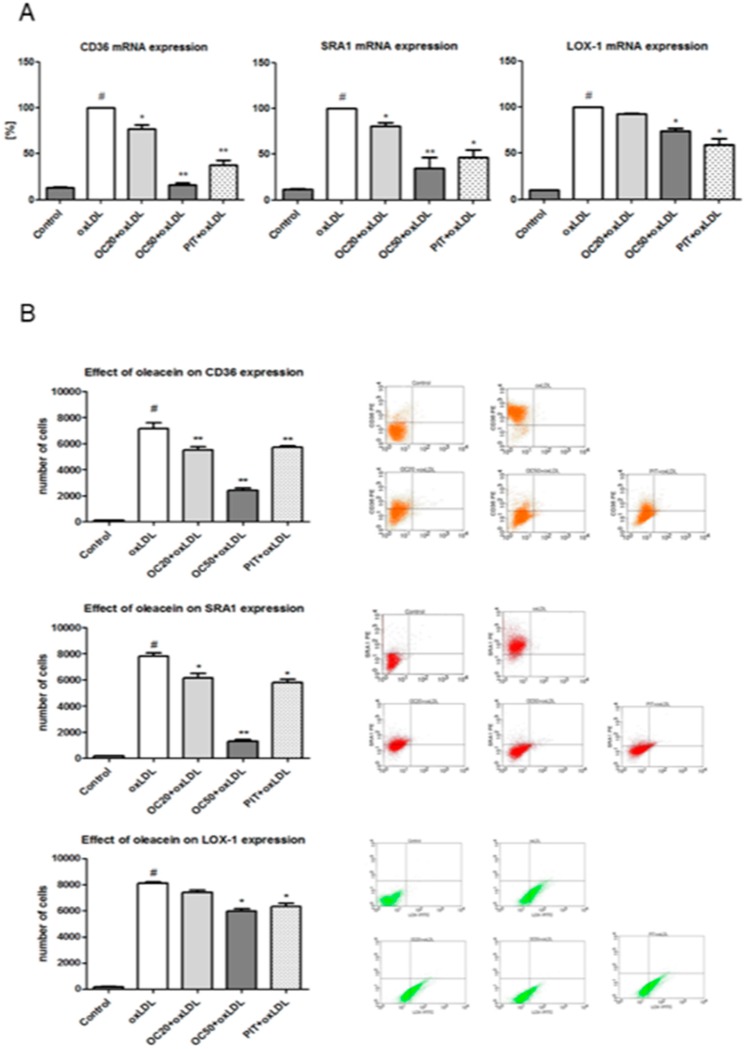
(**A**) Influence of oleacein on CD36 mRNA, SRA1 mRNA and LOX-1 mRNA expression. mRNA levels are shown as arbitrary units normalized to GAPDH expression. Data from 15 experiments ± SEM. Statistical significance * *p* < 0.05, ** *p* < 0.001 compared to the oxLDL-induced macrophages. (**B**) Influence of oleacein on CD36, SRA1 and LOX-1 expression. The results are presented as the number of cells with CD36, SRA1 or LOX-1 expression ± SEM (*n* = 15). Statistical significance * *p* < 0.05, ** *p* < 0.001 compared to the oxLDL-induced macrophages. In dot plots representative results are shown. #—constant relative to which statistics are calculated.

**Figure 4 pharmaceuticals-13-00064-f004:**
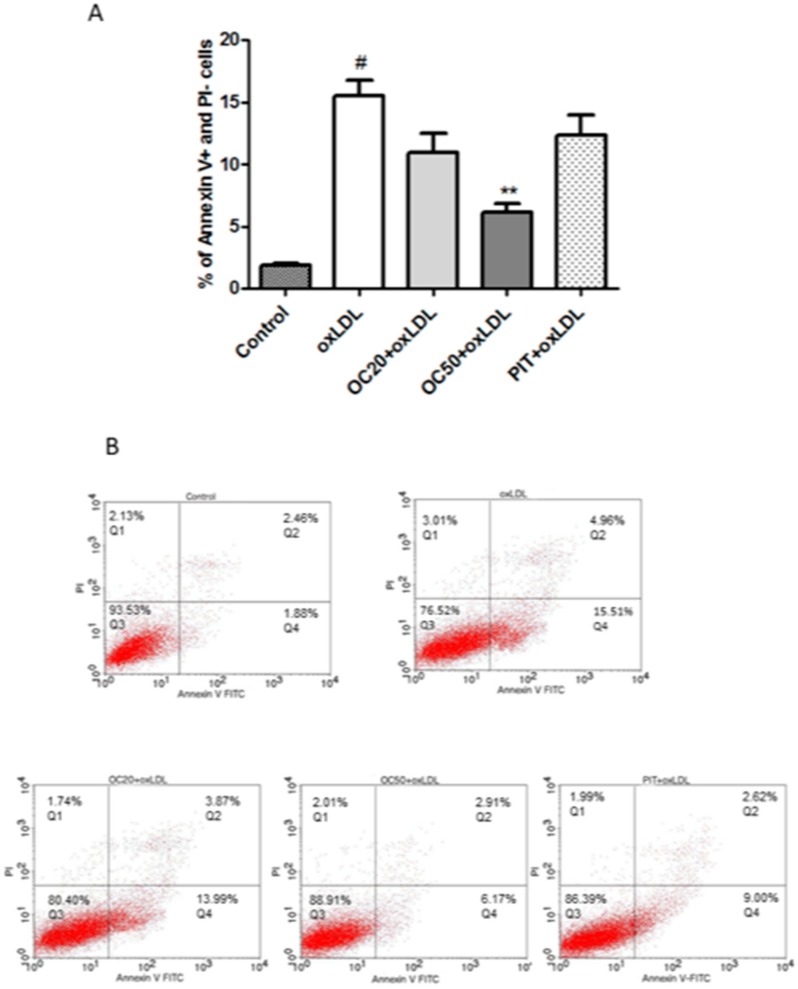
Influence of oleacein on oxLDL-induced early apoptosis. The macrophages were incubated with oxLDL (100 μg/mL) or oxLDL with oleacein (20 μM and 50 μM) for 24 h. (**A**) Percentage of early apoptotic cells (*n* = 12) compared to the oxLDL-induced macrophages. Statistical significance ** *p* < 0.001. #—constant relative to which statistics are calculated. (**B**) Representative results from one of the twelve experiments are shown. Early apoptosis cells with Annexin V^+^/PI^−^ (Q4).

**Figure 5 pharmaceuticals-13-00064-f005:**
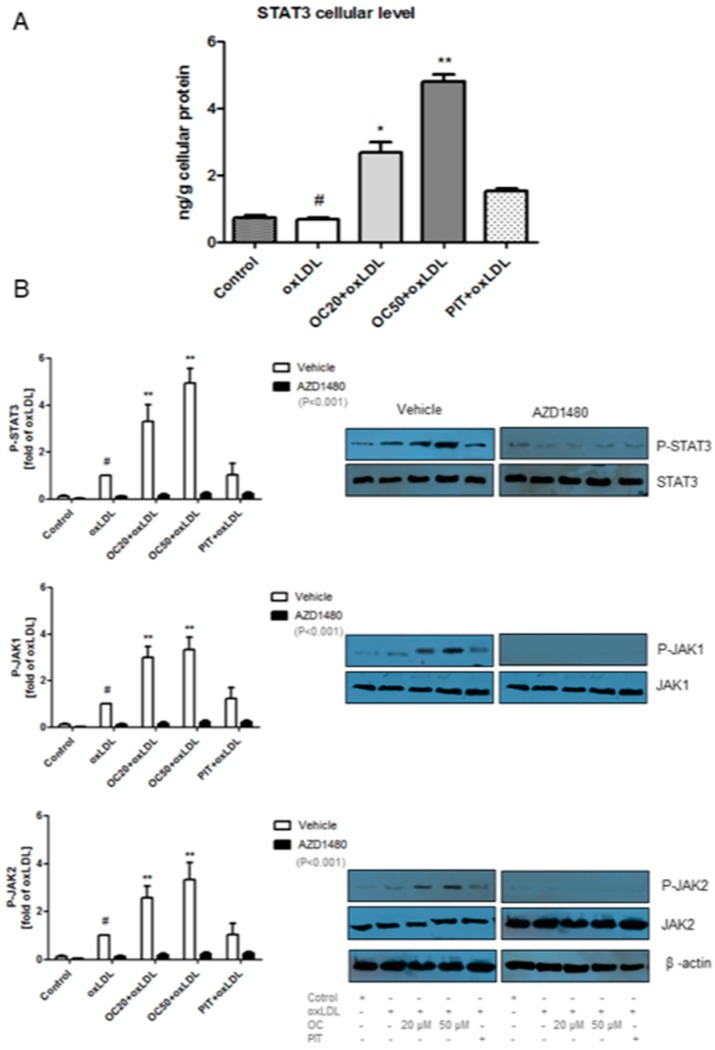
(**A**) Influence of oleacein on STAT3 cellular level by oxLDL-induced macrophages was analysed by ELISA test. The experiment was performed on cells from different donors (*n* = 15). Statistical significance * *p* < 0.05, ** *p* < 0.001 compared oxLDL-induced macrophages. (**B**) Phosphorylation of STAT3 (P-STAT3/STAT3), JAK1 (P-JAK1/JAK) and JAK2 (P-JAK2/JAK2) was analysed by Western Blot and based on total protein. The results were quantified by densitometry. Representative images from one of tree experiments are shown. β-actin was used as an internal control. Vehicle: statistical significance ** *p* < 0.001 compared to the oxLDL-induced macrophages. As an inhibitor of JAK/STAT3 pathway. AZD1480, was used. Statistical significance ** *p* < 0.001 compared to factors without AZD1480.

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
