# Peer review of "Oleacein and Foam Cell Formation in Human Monocyte-Derived Macrophages: A Potential Strategy against Early and Advanced Atherosclerotic Lesions"

_pharmaceuticals, 2020, doi:10.3390/ph13040064_

Round 1

Reviewer 1 Report

The manuscript is focused on pharmacological effects of oleacein on a molecular level and based on different in vitro results which show convincing details regarding the concentration dependent effect of oleacein on human monocytes. The investigations are performed according to high quality standard of research.

Nevertheless, the key question arising from all in vitro and ex vivo experiments is the transfer to the situation in vivo in the human body, esp. the plasma concentration of oleacein available after oral intake of an oleacein containing preparation. Unfortunately, no discussion is focused on that. The authors should refer to that issue with the paper from Agrawal et al. J Funct Foods (2017)36: 84.

Some questions are open and should be discussed:

  1. Is there any chance to obtain a plasma concentration of 50 µM in humans after oral intake of appropriate products?
  2. Please, add some data about the bioavailability of oleacein.
  3. The authors used pitavastatin as positive control – regarding the effects on scavenger receptor expression. To clarify the question of selectivity and specificity of oleacein a polyphenol control would have more importance, e.g. rosmarinic acid or tyrosol. For both substances plasma concentrations are available in literature.

Author Response

We are grateful for the comments and valuable suggestions from Reviewer regarding improvements of the paper. We have addressed the issues raised by the Reviewer point-by-point below and we provided new data as requested by Reviewer. In the current version of manuscript all changes are highlighted.

Nevertheless, the key question arising from all in vitro and ex vivo experiments is the transfer to the situation in vivo in the human body, esp. the plasma concentration of oleacein available after oral intake of an oleacein containing preparation. Unfortunately, no discussion is focused on that. The authors should refer to that issue with the paper from Agrawal et al. J Funct Foods (2017)36: 84.

We would like to thank to the Reviewer for raising this important issue. In the current version of manuscript we added the following paragraph to the Discussion section:

“In the discussion on the potential role of oleacein as a medicinal substance, its bioavailability should be taken into account. It is known that about 70% of oleacein reaches the small intestine in unchanged form and then enters the circulatory system in the form of gluconic acid and glucuronides [47,48]. It should be remembered that inflammatory cells may secrete liposomal β-glucuronidase into the extracellular space and oleacein may be re-released from glucuronides in regions of inflammation [49, 50]. 

The best source of oleacein is unfiltered extra virgin olive oil. Unfortunately, we have several restrictions here. The oleacein content in oils is very different and depends on many factors such as olive tree growing conditions, temperature, insolation, hydration and the method of pressing olive oil. The richest in oleacein are olive oils from the Greek islands and polyphenol-rich unfiltered extra virgin olive oil may not be available to consumers other than Mediterranean countries [48, 51]. It should be noted, that polyphenols are an unstable compound in higher temperature. For this reason, extra virgin olive oil should be consumed in a raw form [52]. In addition, olive oil is rich in fatty acids. Consumption of more than 2 tablespoons of extra virgin olive oil can exceed the daily requirement of adult for fatty acids. Therefore, the daily intake of polyphenols can be too low to induce a therapeutic effect. Due to the above reason, we recommend using oleacein in the form of a pure compound as a dietary supplement or medical drug."

  1. Pinto, J., Paiva-Martins, F., Corona, G., Debnam, ES.., Jose Oruna-Concha, M., Vauzour, D., Gordon, M.H., Spencer, J.P. (2011). Absorption and metabolism of olive oil secoiridoids in the small intestine. Br J Nutr. 105, 1607-18.
  2. Agrawal, K., Melliou, E., Li, X., Pedersen, T.L., Wang, S.C., Magiatis, P., Newman, J.W., Holta R.R. (2017). Oleocanthal-rich extra virgin olive oil demonstrates acute anti-platelet effects in healthy men in a randomized trial. J Funct Foods. 36, 84–93.
  3. Bartholomé, R., Haenen, G., Hollman, C.H., Bast, A., Dagnelie, P.C., Roos, D., Keijer, J., Kroon, P.A., Needs, P.W., Arts, I.C. (2010). Deconjugation kinetics of glucuronidated phase II flavonoid metabolites by beta-glucuronidase from neutrophils. Drug Metab Pharmacokinet. 25, 379-87.
  4. Shimoi, K., Saka, N., Nozawa, R., Sato, M., Amano, I., Nakayama, T., Kinae, N. Deglucuronidation of a flavonoid, luteolin monoglucuronide, during inflammation. (2001). Drug Metab Dispos. 29, 1521-4.
  5. Karkoula, E., Skantzari, A., Melliou, E., Magiatis, P. (2012). Direct measurement of oleocanthal and oleacein levels in olive oil by quantitative (1)H NMR. Establishment of a new index for the characterization of extra virgin olive oils. J Agric Food Chem. 60, 11696-703
  6. Malik, N.S.A., Bradford, J.M. (2008). Recovery and stability of oleuropein and other phenolic compounds during extraction and processing of olive (Olea europaea L.) leaves. J. Food Agric. Environ. 62, 8-13.

  1. Is there any chance to obtain a plasma concentration of 50 µM in humans after oral intake of appropriate products?

Yes, we think there is a chance of getting 50 µM in the blood. Assuming that oleacein is absorbed into the bloodstream at a level of 60-70%, taking 20-30 mg of a pure compound can achieve a concentration of 50 μM in the blood. According to our research, high doses of oleacein are not toxic to cells. Therefore, the use of combination therapy: unfiltered extra virgin olive oil and supplementation with pure compound, may be beneficial for health. Moreover, it should be remembered that oleacein enters the circulatory system in the form of gluconic acid and glucuronides. Inflammatory cells may secrete liposomal β-glucuronidase into the extracellular space. Therefore, oleacein may be re-released from glucuronides in regions of inflammation.

  1. Please, add some data about the bioavailability of oleacein.

According to data from Pinto et al, 2010, 67% of oleacein passes unchanged into the small intestine, where it is absorbed into the circulatory system.

“Incubation of 3,4-DHPEA-EDA and 3,4-DHPEA-EA at pH 2·0 led to the partial hydrolysis of both compounds and a corre- sponding time-dependent increase in HT (Fig. 1). After 4h incubation, about 67 % of 3,4-DHPEA-EDA and 78 % of 3,4- DHPEA-EA (Fig. 1(A) and (B)) remained. The present results show that although some hydrolysis takes place releasing free HT from both 3,4-DHPEA-EDA and 3,4-DHPEA-EA, a large amount of both DHPEA-EDA and 3,4-DHPEA-EA (67 and 78%, respectively, after 4h) remains intact despite pro- longed exposure to postprandial gastric conditions. This is in agreement with previous studies, which suggest that both compounds are also relatively stable at pH environments akin to those in the small intestine with more than 90% of 3,4-DHPEA-EDA and more than 65% of 3,4-DHPEA-EA remaining intact after a 48h incubation at pH 7·4 (378C)(32). These data suggest that both 3,4-DHPEA-EDA and 3,4- DHPEA-EA may be relatively stable during transit through the stomach and the small intestine in vivo. As a consequence, both compounds are likely to arrive at relatively high concentration in the small intestine where they may undergo absorption and metabolism."

Pinto, J., Paiva-Martins, F., Corona, G., Debnam, ES., Jose Oruna-Concha, M., Vauzour, D., Gordon, M.H., Spencer, J.P. (2011). Absorption and metabolism of olive oil secoiridoids in the small intestine. Br J Nutr. 105, 1607-18.

  1. The authors used pitavastatin as positive control – regarding the effects on scavenger receptor expression. To clarify the question of selectivity and specificity of oleacein a polyphenol control would have more importance, e.g. rosmarinic acid or tyrosol. For both substances plasma concentrations are available in literature.

Yes, we know that other phenolic compounds can inhibit the expression of CD36, SRA1 and LOX-1 receptors. However, there is a lot of research on this subject and the issue is well documented. Therefore, as a positive control, we decided to use pitavastatin. The assumption of our research was to compare the activity of oleacein with a synthetic drug. We chose pitavastatin because it is one of the few statins that inhibits the activation of scavenger receptors. We wanted to show that the effect of oleacein is as strong as that of a synthetic drug.

In the current version of manuscript, the conclusions are corrected

“Our results suggested that oleacein may reduce foam cell formation from human macrophages by decreasing expression of scavenger receptors such as CD36, SRA1 and LOX-1. Moreover, oleacein can protect against foam cell formation by switch macrophages from pro-inflammatory type M1 to anti-inflammatory type M2 via JAK/STAT3 pathway activation. In addition, oleacein may inhibited oxLDL-induced early apoptosis of macrophages. This beneficial effect is associated with the multifaceted, so-called pleiotropic activity of oleacein.  Therefore, we suggest oleacein as a substance for use in strategies against early and advanced atherosclerosis. Noteworthy is the fact that the source of oleacein may be the leaves of Ligustrum vulgare (Oleaceae), which is a common plant, not only in Mediterranean countries.

Nevertheless, these studies still require confirmation on an animal model. However, they indicate that in the future oleacein may play a significant role in preventing heart disease.”

Reviewer 2 Report

The authors investigate the role of oleacein in the prevention of foam cell formation in vitro. Endpoints include lipid deposition, expression of CD36, SRA1 and LOX-1, apoptosis, expression of JAK/STAT3 and ACAT1. The authors find a benefit of oleacetic with regards to all these endpoints. 

I do not have major comments, except for the fact that it would be nice to see validation in a in vivo model. 

Author Response

We are grateful for the valuable comments from Reviewer.

In vivo studies are planned. The Apoe−/− mice model is a suitable model to study cardiovascular diseases. Therefore, we intend to use Apoe-/- mice to study oleacein in the context of atherosclerosis. Nevertheless, preliminary studies confirm that oleacein inhibits lipid accumulation by mouse's hepatocytes, which is probably associated with scavenger receptors expression [1].

  1. Lombardo, G.E., Lepore, S.M., Morittu, V.M., Arcidiacono, B., Colica, C., Procopio, A., Maggisano, V., Bulotta, S., Costa, N., Mignogna, C., Britti, D., Brunetti, A., Russo, D., Celano, M. (2018). Effects of Oleacein on High-Fat Diet-Dependent Steatosis, Weight Gain, and Insulin Resistance in Mice. Front Endocrinol (Lausanne). 19, 116.

Round 2

Reviewer 1 Report

My remarks and questions were considered. I am satisfied with the changes.